# ROYAL SOCIETY
# OPEN SCIENCE

molecular computing

finite state automata, approximate majority, abstract chemical reaction networks

**Author for correspondence:**
Matthew R. Lakin
e-mail: mlakin@cs.unm.edu

# Robust finite automata in stochastic chemical reaction networks

David Arredondo[3] and Matthew R. Lakin[1,2,3]

[1]Department of Computer Science, and [2]Department of Chemical and Biological Engineering, and [3]Center for Biomedical Engineering, University of New Mexico, Albuquerque, NM 87131, USA

MRL, 0000-0002-8516-4789

Finite-state automata (FSA) are simple computational devices that can nevertheless illustrate interesting behaviours. We propose that FSA can be employed as control circuits for engineered stochastic biological and biomolecular systems. We present an implementation of FSA using counts of chemical species in the range of hundreds to thousands, which is relevant for the counts of many key molecules such as mRNAs in prokaryotic cells. The challenge here is to ensure a robust representation of the current state in the face of stochastic noise. We achieve this by using a multistable approximate majority algorithm to stabilize and store the current state of the system. Arbitrary finite state machines can thus be compiled into robust stochastic chemical automata. We present two variants: one that consumes its input signals to initiate state transitions and one that does not. We characterize the state change dynamics of these systems and demonstrate their application to solve the four-bit binary square root problem. Our work lays the foundation for the use of chemical automata as control circuits in bioengineered systems and biorobotics.

## 1. Introduction

A key goal in the field of molecular programming is to program bioinspired behaviours in engineered biomolecular systems. Even single-celled microorganisms are capable of complex behaviours [1] and many of these can be thought of in terms of the system moving between discrete internal states in response to stimuli, with those states driving subsequent responses to stimuli [2]. In biological cells, discrete state transitions control checkpointing within the cell cycle to activate cell division [3] and the progression from stem cells to differentiated cell types [4]. In all of these systems, a key aspect is the imposition of a discrete state change. This must be coordinated throughout the physical extent of the cell so that the state change is committed to by the entire distributed system, in the face of the noise

inherent in stochastic chemical systems containing of the order of hundreds to thousands of copies of individual chemical species. Here, we present designs for biomolecular devices that robustly implement such finite state automata (FSA) using stochastic chemical reaction networks.

Previous theoretical work has shown that abstract chemical reaction networks are Turing-powerful, albeit with an arbitrarily small probability of error [5]. In addition, previous work has shown that polymer-forming DNA strand displacement networks can exactly implement stack machines [6,7], which are equivalent to Turing machines. This approach uses polymer-based 'stack' models of unbounded length to store information while using a finite state control program to control state transitions based on memory content and the current state. Thus, while the emulation is exact in this case, an infinite chemical reaction network is required. In addition, to ensure correctness, only a single molecule of each stack type can be present and only one state molecule can be present. This produces a consistent representation of the state at all times but is infeasible in a practical implementation. Increasing the number of copies of these molecules in the system raises the possibility of errors, and various schemes have been proposed to correct these errors, which typically involve detecting the error and replaying the computation to try again [8]. Thus, while of significant theoretical interest, Turing-complete molecular programming systems are not currently a practical route to implement bioinspired control programs for biomolecular systems.

Leaving the stack component aside, the finite state control program viewed on its own can nevertheless produce interesting behaviours that may be more amenable to practical implementation. FSA are abstract computational systems that move between states drawn from a finite set, with transitions chosen based on the current state and an observed input signal. We conjecture that these are sufficient to implement interesting behaviours in molecular systems, even though they are strictly weaker computationally than Turing machines. In this paper, we use abstract chemical reaction networks (CRNs) to implement finite automata capable of robust state changes under a stochastic chemical reaction network semantics, and in the biologically relevant range of hundreds to thousands of copies of each species. Previous work has shown that arbitrary CRNs can be compiled into DNA strand displacement networks [9], thereby providing a path to practical implementations of our designs in wet chemistry. Our designs use a multistable extension of the well-known approximate majority switch [10], which has previously been implemented in DNA [11], to store the current state in a robust manner. In conjunction with this memory mechanism, we use a simple 'flipping' scheme and a finite control program to trigger state transitions as a function of the current state and the observed sequence of inputs provided to the system, thereby mimicking the behaviour of a particular theoretical finite state automaton. We characterize the robustness of our flipping mechanism in terms of the corresponding rate constant values and provide a compilation scheme to translate FSA directly into abstract CRNs. Our CRN implementation of deterministic finite automata could be a powerful tool for molecular programmers to build complex sensing and decision-making programs.

## 2. Results

Chemical reaction networks can be viewed as distributed computing systems, as each individual molecule acts independently according to the reactions in which it can participate. The stochastic nature of chemical reactions gives rise to noise, especially in systems with low molecular counts. A robust chemical representation of the abstract system state is required to reliably implement systems such as our chemical FSA. Here, we present an approach to achieve this using the approximate majority (AM) algorithm for distributed consensus [10]. Originally developed to enable a distributed sensor network to reliably converge to a state representing the initial majority consensus, similar networks have been found natively in biological systems, for example to coordinate switching events in the cell cycle [3,12]. Furthermore, the AM reaction has been realized in DNA using strand displacement reactions [11]. Here, we use a multistable extension of the AM algorithm to enable robust representation of the state of a finite state automaton in a stochastic chemical system. This is achieved because the chemical species representing the states are the inputs to an AM module, as outlined below. This ensures global consensus on the current state of the FSA, even in the face of stochastic noise.

We have defined a model for deterministic FSA represented as chemical reaction networks and show that arbitrary deterministic FSA can be compiled into this representation. The system is composed of multiple CRN modules representing the different functions necessary for the FSA to function including input processing, state transitioning and state stabilization. The minimal system is designed

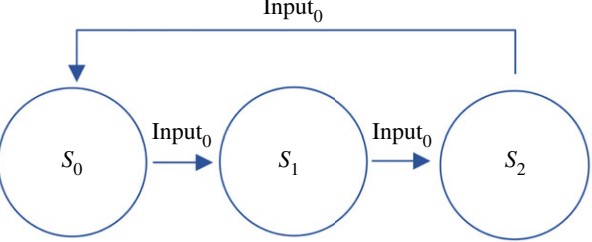

**Figure 1.** State diagram for three-state cycle example FSA. Sequential presentations of $Input_0$ transition the FSA from state $S_0$ to $S_1$, then to $S_2$, and then back to $S_0$ to complete the cycle.

Approximate majority module:

$$S_0 + S_1 \xrightarrow{k_{AM}} B + B \qquad\qquad S_0 + B \xrightarrow{k_{AM}} S_0 + S_0$$

$$S_0 + S_2 \xrightarrow{k_{AM}} B + B \qquad\qquad S_1 + B \xrightarrow{k_{AM}} S_1 + S_1$$

$$S_0 + S_2 \xrightarrow{k_{AM}} B + B \qquad\qquad S_2 + B \xrightarrow{k_{AM}} S_2 + S_2$$

Transition mapping module:

$$Input_0 + S_0 \xrightarrow{k_{Input}} Flip_{0\rightarrow1} + S_0 \qquad Input_0 + S_1 \xrightarrow{k_{Input}} Flip_{1\rightarrow2} + S_1$$

$$Input_0 + S_2 \xrightarrow{k_{Input}} Flip_{2\rightarrow0} + S_2$$

Transition module:

$$Flip_{0\rightarrow1} + S_0 \xrightarrow{k_{Flip}} S_1 \qquad\qquad Flip_{1\rightarrow2} + S_1 \xrightarrow{k_{Flip}} S_2$$

$$Flip_{2\rightarrow0} + S_2 \xrightarrow{k_{Flip}} S_0$$

Flip decay module:

$$Flip_{0\rightarrow1} \xrightarrow{k_{FlipDecay}} \varnothing \qquad\qquad Flip_{1\rightarrow2} \xrightarrow{k_{FlipDecay}} \varnothing$$

$$Flip_{2\rightarrow0} \xrightarrow{k_{FlipDecay}} \varnothing$$

**Figure 2.** Example implementation of the three-state cycle FSA from figure 1 as a stochastic CRN. This version of the FSA CRN consumes its input signals to initiate state transitions.

to receive inputs as an instantaneous addition of an input species and to consume these inputs; however, the ultimate goal is to interface chemical FSA with real biochemical systems. We therefore also introduce a module that uses a catalyst to convert a buffer of inactive input to active input and test this on inputs following a Boolean model of gene expression. In this model, the gene product acts as the catalytic input, effectively producing a consumable spike of active input on the rising edge of the square wave of the input signal.

## 2.1. Approximate majority algorithm for finite state automata

To demonstrate the core modules and behaviour of the deterministic chemical FSA, we have studied a simple system with three states ($S_0$, $S_1$ and $S_2$) and one input ($Input_0$). Addition of the input will trigger a transition from one state to the next in a three-state cycle, as shown in figure 1. This provides a minimal system to explain our design and also to study the dynamics of state transitions, including possible failure modes. Its CRN implementation, outlined in figure 2, comprises three modules: the approximate majority module, the transition mapping module and the transition module.

The approximate majority module is crucial for the stability of the system. It converts distinct state species $S_i$ to the 'undecided' species $B$, which is then recruited back into a state species with higher probability if that species is present in a larger quantity within the system. This ensures that, upon perturbation, the system will converge to one of three stable states, in which all state molecules have

been converted to $S_0$, $S_1$ or $S_2$. These represent the three abstract states of the underlying FSA. In general, the AM module can be generalized to choose between $n$ states $S_0, \ldots, S_{n-1}$ via a similar process, implemented using the reactions:

— $S_i + S_j \xrightarrow{k_{AM}} B + B$ for all $i, j \in \{0, \ldots, n-1\}$ for which $i < j$; and
— $S_i + B \xrightarrow{k_{AM}} S_i + S_i$ for all $i \in \{0, \ldots, n-1\}$.

Thus, if there are $n$ distinct states to choose between, the number of species scales as $O(n)$ and the number of reactions scales as $O(n^2)$. We require that the rate constants for the AM reactions are the same ($k_{AM}$). We investigate rate constant values further below.

## 2.2. Transition mapping reactions with input consumption

We now present our design for implementing state transitions via consumption of an instantaneous spike of input species into the system. Consumption of these input species drives the state transition. This design corresponds to the transition mapping module, the transition module and the flip decay module from figure 2.

First, the transition mapping module specifies the possible state transitions by pairing up the state species and input species as the reaction arguments. In each reaction, the state species catalyses conversion of the input species into a 'flip' species that represents the desired state transition. There is one flip species per transition, e.g. Flip$_{0\to1}$, which represents the transition from state $S_0$ to state $S_1$. Second, the transition module actually implements the state transitions: the flip species are consumed in order to flip the corresponding 'initial' state species into the resulting 'final' state species. This two-step process is required because we wish to robustly implement *arbitrary* FSA architectures in our CRNs, which includes loops and self-edges.

Figure 3 shows the result of stochastic simulations with 100 state molecules, both with and without the 'flip' species, to illustrate their importance in enabling robust state transitions. Simulations were performed using Gillespie's algorithm [13]. In the case without the 'flip' species, the corresponding reactions from the transition mapping module and transition module are compressed into single reactions (note that the products of the transition mapping module reactions match the reactants of the transition module reactions). In other words, in the 'no flip' case, the transition mapping and transition modules from figure 2 are replaced with the following three reactions:

$$\text{Input}_0 + S_0 \xrightarrow{k_{Input}} S_1 \quad \text{Input}_0 + S_1 \xrightarrow{k_{Input}} S_2.$$

$$\text{Input}_0 + S_2 \xrightarrow{k_{Input}} S_0$$

Three spikes of 80 molecules of Input$_0$ are provided at $t = 0.25$, 0.5 and 0.75. In figure 3a, we see that the absence of the flip species step leads to significantly more noise in the system after the input spikes have been applied, which could lead to one of two major failure modes in a state transition: (i) failing to switch out of the initial state or (ii) 'skipping' one or more states by effectively carrying out multiple partial state transitions that the AM module resolves into an incorrect state. By contrast, figure 3b shows that the inclusion of the flip species stabilizes the system in the correct 'next' state after each input presentation. To provide statistical evidence of the need for the flip species, figure 3c,d presents the mean and standard deviation species counts over 100 simulations of the systems that produced the representative traces from figure 3a,b. In the presence of the flip species (figure 3d), we see tight error bars around the expected sequence of state transitions, whereas in the absence of the flip species (figure 3c), we see wide error bars as the incorrect transitions lead to significant variation in the means of the state species counts across all traces.

Failing to transition occurs when the consumed input cannot produce enough flip molecules to convert more than 50% of the state species before the AM reactions reassert themselves and restabilize the system in the initial state. The second attempted transition from figure 3a, at time $t = 0.5$, is an example of a failed transition. This can occur if insufficient input is provided: in our simulations, we use 80 input molecules compared with 100 state molecules to ensure that we exceed this threshold. Failing to transition is a particular problem in systems where the state being transitioned to also has an edge back to the state being transitioned from; in this situation, any noisy creation of the second flip species will reinforce the pressure on the system not to leave the initial state in the first place. Thus, our approach is particularly useful for implementing FSA architectures involving loops, which are common as they enable a finite state system to process input sequences of unbounded size. In

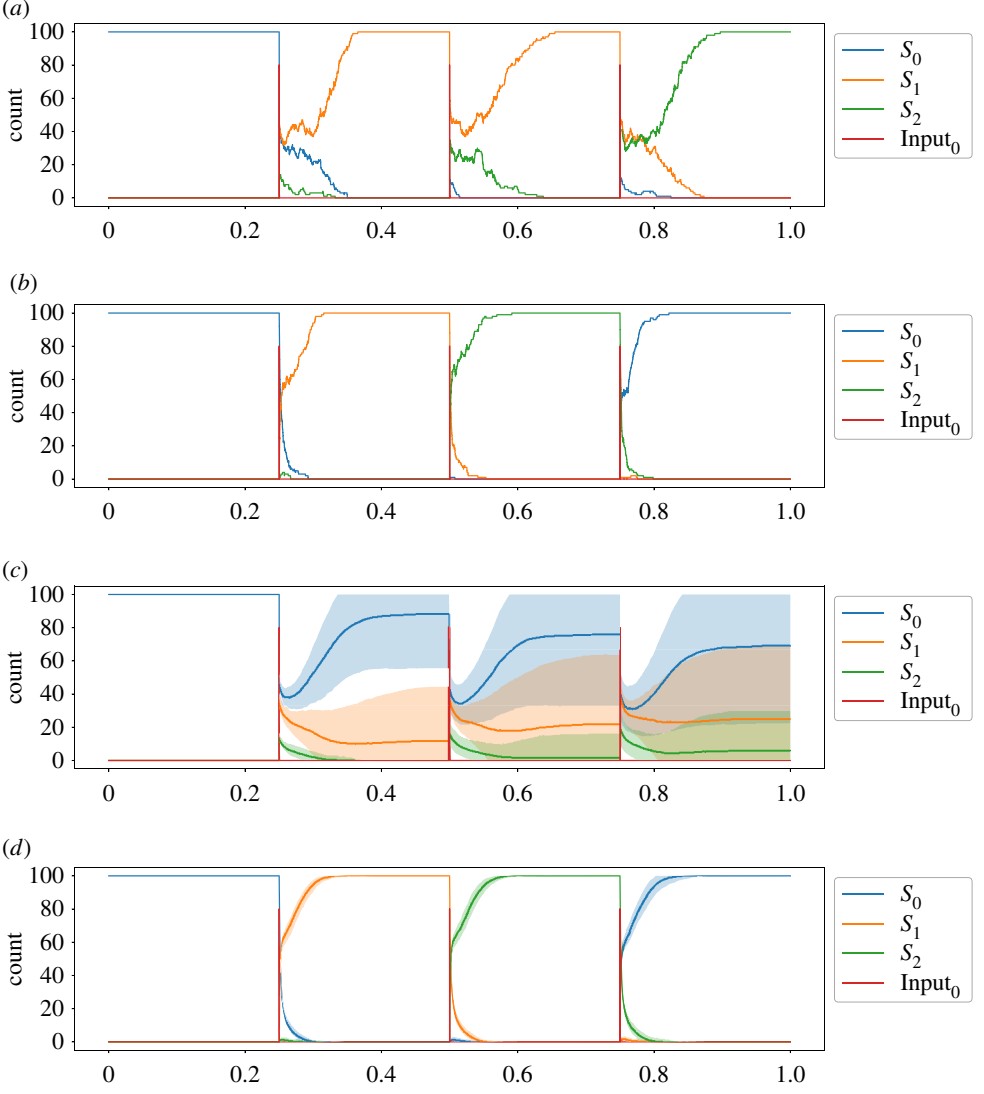

**Figure 3.** Demonstration of three-state cycle CRN, with and without the flip species in the transition mapping module. The rate constants used are $k_{AM} = 1$, $k_{Flip} = 10$, $k_{Input} = 100$ and $k_{FlipDecay} = 1$. Three spikes of 80 molecules of $Input_0$ are provided at $t = 0.25$, 0.5 and 0.75. (*a*) Example state transition activated by consumable input, *without* flip species, (*b*) example state transition activated by consumable input, *with* flip species, (*c*) average species count for 100 traces of the system shown in (*a*), *without* flip species. Shaded area represents 1 s.d. above and below the mean, truncated to [0, 100], (*d*) average species count for 100 traces of the system shown in (*b*), *with* flip species. Shaded area represents 1 s.d. above and below the mean, truncated to [0, 100].

addition, our system is robust to perturbations due to the introduction of small amounts of input, as could occur in a noisy environment, as their effect is nullified by the AM reactions.

Skipping states by making multiple transitions on one presentation input can occur because FSAs define a transition for each input from each state. Thus, when an input triggers a transition from state $S_i$ to $S_j$, there will also be a valid transition from $S_j$ to another state, $S_k$ say, for the same input. In our FSA CRN implementation, this means that once the state species $S_j$ begins to be produced, it can begin to react with the input to produce the $Flip_{j \to k}$ species, and so on. However, there is only a finite amount of input to consume and the population of flip species produced should be dominated by those for the first transition in the chain from the current state, due to being catalysed by the initial state species in the transition mapping module. This fact, in conjunction with well-chosen kinetic rate constants, enables us to implement reliable state transitions in our FSA CRNs. (Indeed, no example of state skipping is observed in figure 3*a*, and we observe this primarily in conditions with excess amounts of input present, e.g. in figure 4.) Here and henceforth, we use the rate constants $k_{AM} = 1$, $k_{Flip} = 10$ and $k_{Input} = 100$, unless stated otherwise. These rate constants define a separation of

timescales between the reactions that process inputs into flip species, the reactions that consume those flip species to initiate a state transition and the AM reactions that stabilize the new state. We discuss the rationale for these choices and explore robustness of the system to changes in these rate constants, in the following section, where we carry out parameter sweeps to justify our rate constant choices.

The final module shown in figure 2 is the flip decay module. In the absence of this module, the populations of flip species may not return to zero after the state transition. This means that memory of the previous state transition is preserved in the system, which would influence the kinetics of subsequent state transitions. This situation could arise either due to there being an excessively large input spike and/or due to the contribution of the AM reaction to the reaction flipping, meaning that not all of the flip species is actually required to complete the state transition. The flip species must be returned to a memoryless state after each transition, therefore, we add the degradation reactions shown in the flip decay module, one per possible state transition in the system. Unless otherwise stated, in subsequent simulations we set the rate constant of this reaction to $k_{FlipDecay} = 1$.

## 2.3. Characterization of rate parameters for reliable state transitions

In this section, we characterize the reliability of the switching behaviour in our stochastic AM systems. Note that in this simulation and in the previous examples, we do not define accepting states and the initial state is arbitrary; for this characterization, we are only concerned with the correctness of individual state transitions. As outlined above, there are two major failure modes in a state transition: failing to switch out of the initial state or 'skipping' one or more states by carrying out multiple partial state transitions in one go instead of just one. To study the effects of modifying the rate constants on the reliability of state transitions, we characterized the correctness of state transitions in several generalized versions of the FSA CRN from figure 2 with different cycle lengths. Specifically, we studied cycles containing two, three and 20 states.

As per the FSA CRN, an added $Input_0$ molecule can, in principle, react with any state species and be converted to a $Flip_{i \to j}$ species according to the transition mapping of the FSA, and the total amount of $Flip_{i \to j}$ species created will be equal to the number of active $Input_0$ species given to the system. Therefore, the key question is the quantities and proportions in which the various $Flip_{i \to j}$ species are created, and how this interacts with the AM-based state restoration mechanism.

In the limit when $k_{Input} \gg k_{Flip} \gg k_{AM}$, we can make the quasi-steady state assumption. Under this assumption, if the system starts in state $S_i$ then all $Input_0$ molecules would first be converted to the flip species $Flip_{i \to j}$ corresponding to the correct transition from state $S_i$ given the input signal $Input_0$. Then, the flip species would all be consumed to convert $S_i$ state molecules into $S_j$. Finally, the AM reactions would fire, restoring all state species to whichever of $S_i$ and $S_j$ is in the majority at that point in time. This analysis highlights that the system will fail to switch the state if the amount of $Input_0$ provided is less than 50% of the number of state species; even if everything else is ideal, *too little* input will never be able to convert enough state species to switch to the next state. Another failure mode may be caused by *too much* input, which may be left over after a sufficient amount has been converted to the desired $Flip_{i \to j}$ species, which could contribute to skipping states as outlined below.

The quasi-steady state assumption is not entirely realistic, however, meaning that in reality the reactions of the transition mapping module, transition module and approximate majority module will be interleaved temporally. With a rate constant $k_{Input}$ closer to $k_{Flip}$, the resulting $Flip_{i \to j}$ species can immediately interact with an $S_i$ state species, converting it to $S_j$. This means there can simultaneously exist $S_i$ and $S_j$ in the system, which will compete to convert the remaining active $Input_0$ molecules into different flip molecules. Because the conversion of $Input_0$ to $Flip_{i \to j}$ is catalysed by $S_i$, our construction relies on a significant amount of $Input_0$ being converted and depleted before the resulting $S_j$ can compete to activate a second state transition that would result in state $S_j$ being 'skipped'. This effect is amplified if an excess of input is presented, or if the rate constants are far enough outside the optimal region, causing the system to make multiple partial transitions and stabilize in some downstream state.

With the above analysis in mind, we characterized the fraction of correct state transitions observed in the $n$-state cycle FSA with consumable inputs, for $n = 2$, 3 and 20. We fixed the rate constants $k_{AM} = 1$ and $k_{FlipDecay} = 1$ and characterized the remaining parameters: the rate constants $k_{Input}$ and $k_{Flip}$, as well as the amount of $Input_0$ added in each input presentation ($N_{Input}$). We chose this section of the parameter space as it covers rate constants up to two orders of magnitude different from the fixed rate $k_{AM} = 1$. We characterized this system in stochastic simulations containing 100 state molecules, which illustrates its behaviour with relatively small molecular counts that are on the

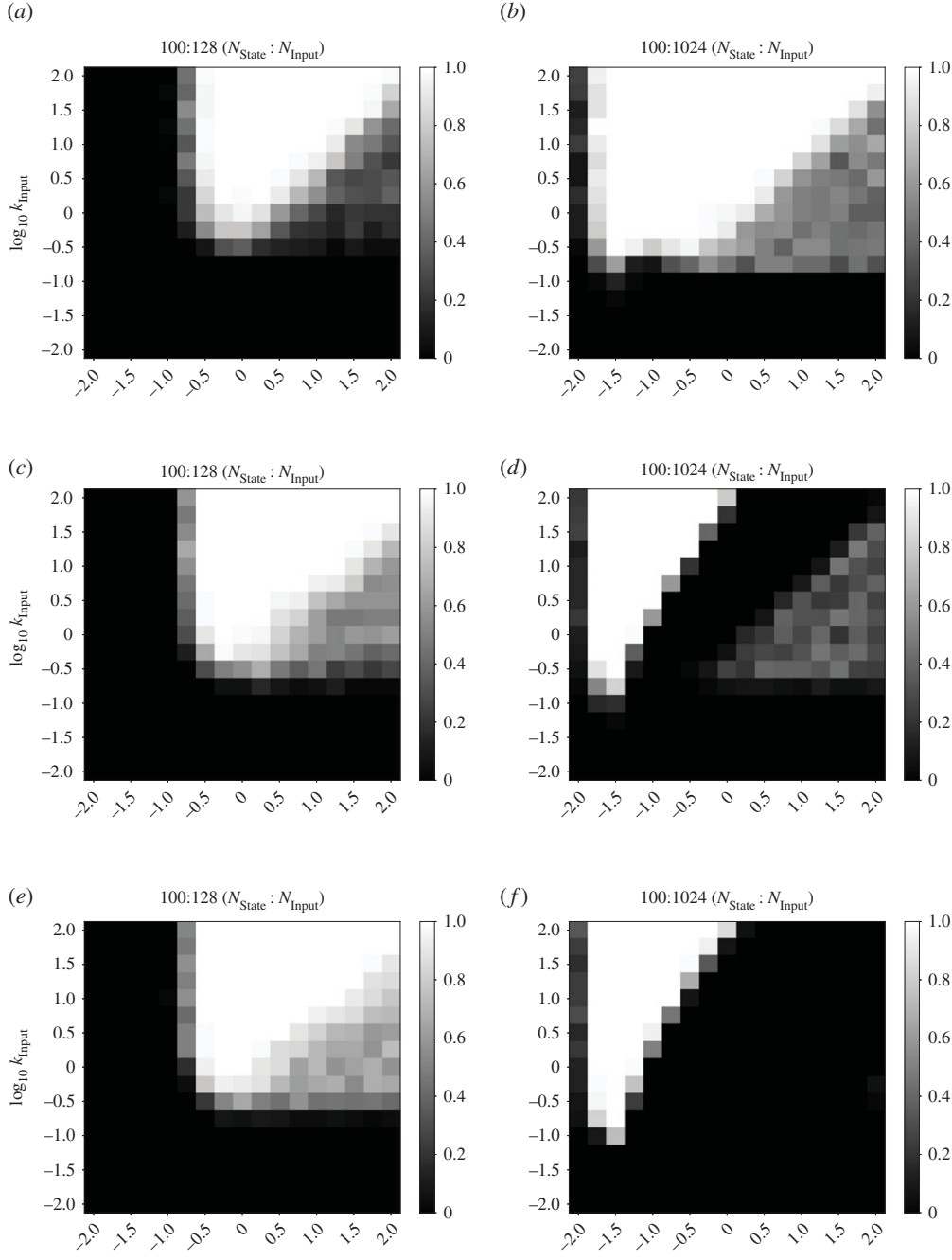

**Figure 4.** Summary of rate characterization data for $n$-state cycle FSA CRNs. Heatmaps show fraction of correct transitions for $n$-state cycle systems with different cycle lengths and different amounts of consumable input presented ($N_{Input}$). For each pixel of the heatmap, five simulations were performed with 10 consecutive presentations of $Input_0$ evenly spaced per trace. Correctness was measured as the total number of successful state transitions divided by the number attempted. A white or black pixel indicates 100% or 0% correctness, respectively. Otherwise, cells are shaded with a greyscale value equal to their correctness. The two axes of the heatmap represent $k_{Input}$ and $k_{Flip}$ on logarithmic scales. The other rate constants were: $k_{AM} = 1$ and $k_{FlipDecay} = 1$. (a) Cycle length = 2, $N_{Input} = 128$, (b) cycle length = 2, $N_{Input} = 1024$, (c) cycle length = 3, $N_{Input} = 128$, (d) cycle length = 3, $N_{Input} = 1024$, (e) cycle length = 20, $N_{Input} = 128$ and (f) cycle length = 20, $N_{Input} = 1024$.

biologically relevant scale of prokaryotes such as *Escherichia coli*. Key results from our stochastic simulations are summarized in heatmap form in figure 4. Figure 4a,b shows results for a two-state cycle with $N_{Input} = 128$ and 1024, respectively. Similarly, figure 4c,d reports results for a three-state cycle and figure 4e,f reports results for a 20-state cycle. Further data for these three systems are presented in figures 5–7.

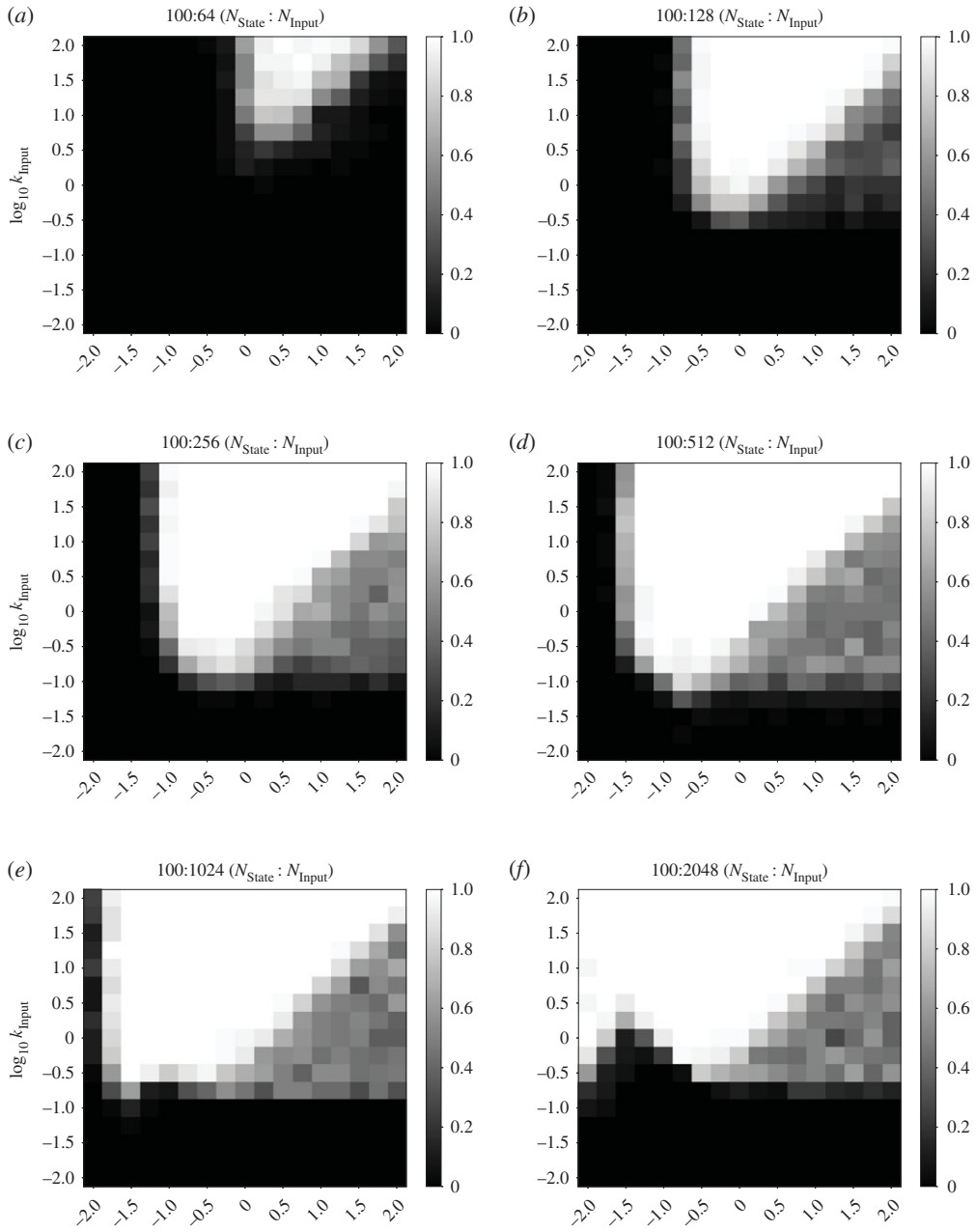

**Figure 5.** Full rate characterization data for the two-state cycle FSA CRN. Heatmaps show fraction of correct transitions with different amounts of consumable input presented ($N_{\text{Input}}$). For each pixel of the heatmap, five simulations were performed with 10 consecutive presentations of $\text{Input}_0$ evenly spaced per trace. Correctness was measured as the total number of successful state transitions divided by the number attempted. A white or black pixel indicates 100% or 0% correctness respectively. Otherwise, cells are shaded with a greyscale value equal to their correctness. The two axes of the heatmap represent $k_{\text{Input}}$ and $k_{\text{Flip}}$ on logarithmic scales. The other rate constants were: $k_{AM} = 1$ and $k_{\text{FlipDecay}} = 1$. (a) Cycle length = 2, $N_{\text{Input}} = 64$, (b) cycle length = 2, $N_{\text{Input}} = 128$, (c) cycle length = 2, $N_{\text{Input}} = 256$, (d) cycle length = 2, $N_{\text{Input}} = 512$, (e) cycle length = 2, $N_{\text{Input}} = 1024$ and (f) cycle length = 2, $N_{\text{Input}} = 2048$.

For each parameter combination, 50 state transitions were simulated in total over five simulations of 10 transitions each. The fraction that correctly transitioned to the next state was calculated in each case. Correct state transitions were detected by averaging the species counts from the middle 50% of time points occurring between two perturbations. For a given time point, if the count of the relevant state species is greater than 80% of the total number of state species (which in this case means 80 copies), then we say the system is in the corresponding state. We observe generally higher correctness percentages for higher values of the $k_{\text{Input}}$ parameter, which matches our intuition that fast conversion

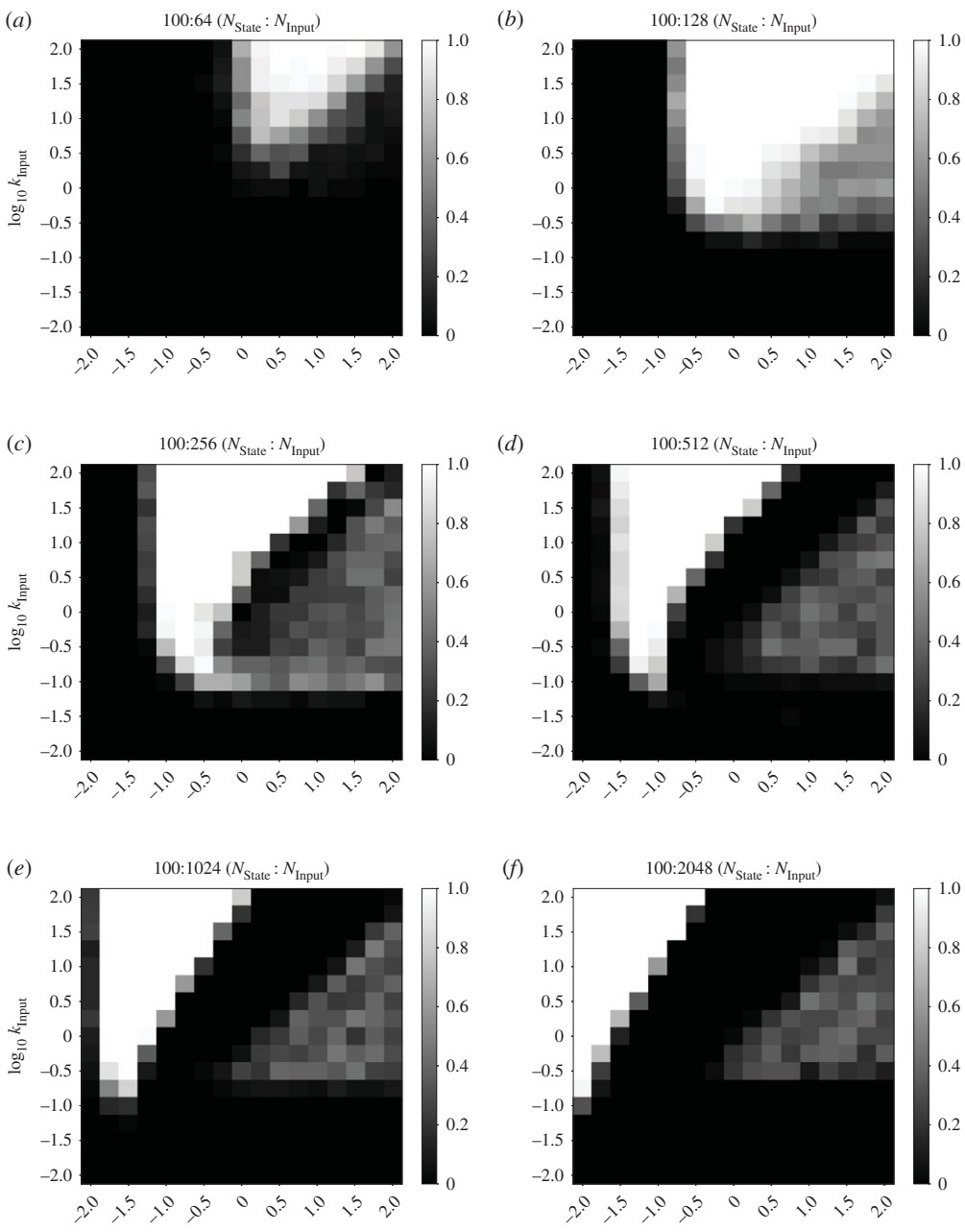

**Figure 6.** Full rate characterization data for the three-state cycle FSA CRN. Heatmaps show fraction of correct transitions with different amounts of consumable input presented ($N_{Input}$). For each pixel of the heatmap, five simulations were performed with 10 consecutive presentations of $Input_0$ evenly spaced per trace. Correctness was measured as the total number of successful state transitions divided by the number attempted. A white or black pixel indicates 100% or 0% correctness respectively. Otherwise, cells are shaded with a greyscale value equal to their correctness. The two axes of the heatmap represent $k_{Input}$ and $k_{Flip}$ on logarithmic scales. The other rate constants were: $k_{AM} = 1$ and $k_{FlipDecay} = 1$. (a) Cycle length = 3, $N_{Input} = 64$, (b) cycle length = 3, $N_{Input} = 128$, (c) cycle length = 3, $N_{Input} = 256$, (d) cycle length = 3, $N_{Input} = 512$, (e) cycle length = 3, $N_{Input} = 1024$ and (f) cycle length = 3, $N_{Input} = 2048$.

of input into flip species is an important component of this system. Furthermore, smaller quantities of input, such as the 64 molecules shown in figures 5–7 produce a smaller zone of parameter space with high correctness. Again, this is unsurprising given that at least 50% of the state species must be flipped to produce a state transition at all, and 64 molecules is not far above this threshold. This zone of high correctness probability grows as the amount of input is added. Large amounts of excess input causes this zone of high correctness probability to shift towards lower $k_{Flip}$ values; this is probably due to the fact that we fixed $k_{FlipDecay} = 1$, meaning that lower $k_{Flip}$ values allow more of the excess

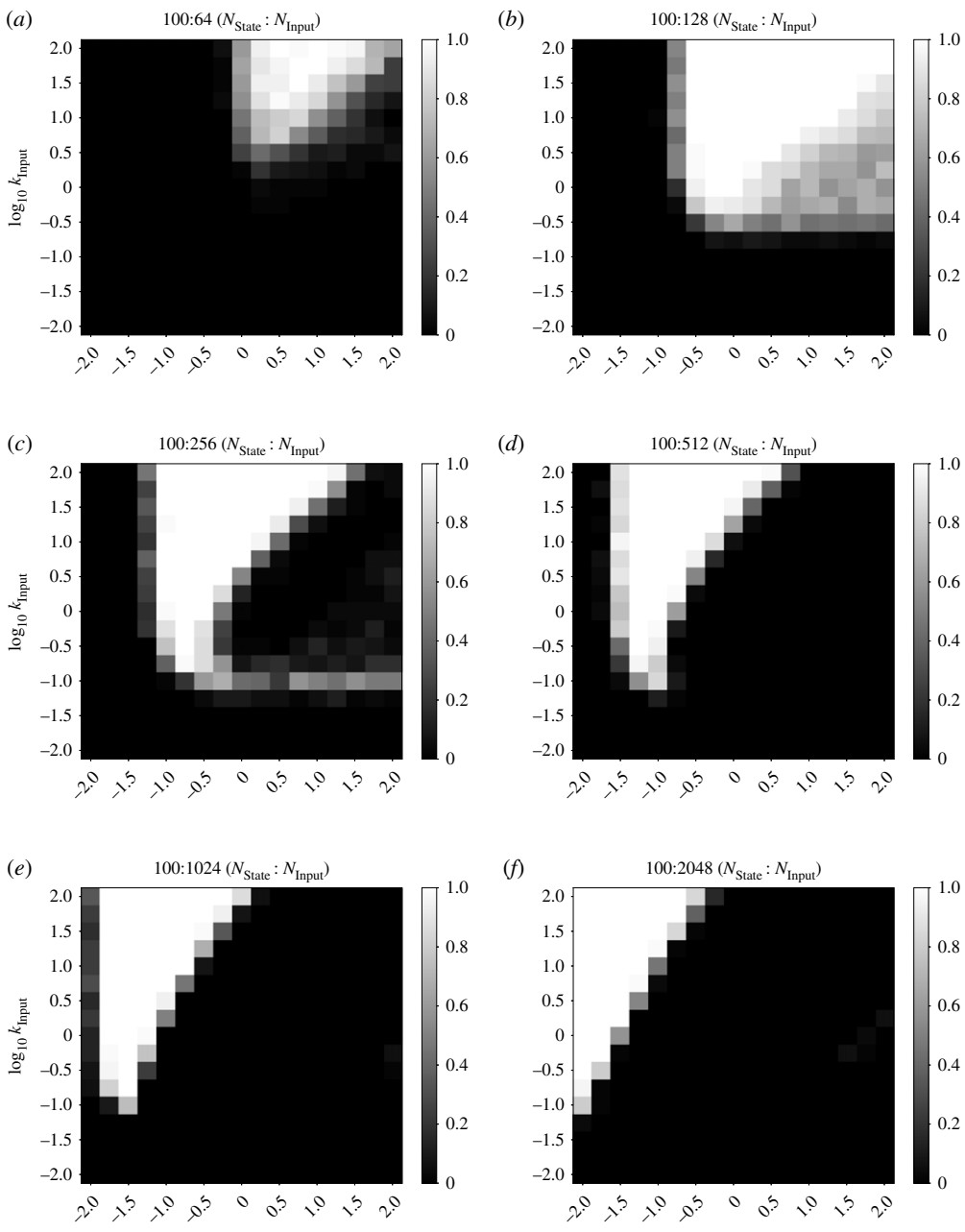

**Figure 7.** Full rate characterization data for the 20-state cycle FSA CRN. Heatmaps show fraction of correct transitions with different amounts of consumable input presented ($N_{Input}$). For each pixel of the heatmap, five simulations were performed with 10 consecutive presentations of $Input_0$ evenly spaced per trace. Correctness was measured as the total number of successful state transitions divided by the number attempted. A white or black pixel indicates 100% or 0% correctness respectively. Otherwise, cells are shaded with a greyscale value equal to their correctness. The two axes of the heatmap represent $k_{Input}$ and $k_{Flip}$ on logarithmic scales. The other rate constants were: $k_{AM} = 1$ and $k_{FlipDecay} = 1$. (a) Cycle length = 20, $N_{Input} = 64$, (b) cycle length = 20, $N_{Input} = 128$, (c) cycle length = 20, $N_{Input} = 256$, (d) cycle length = 20, $N_{Input} = 512$, (e) cycle length = 20, $N_{Input} = 1024$ and (f) cycle length = 20, $N_{Input} = 2048$.

flip species to decay before they can contribute to erroneous state switching reactions. In all cases, the zone at the bottom of the plots, for the lowest values of $k_{Input}$, produces essentially no successful transitions; probably because the production of flip species is so slow that the system cannot transition quickly enough to overcome the AM reactions.

Also of interest is the region of moderately successful state transitions on the right-hand side of these plots, roughly in the region where $k_{Flip} > k_{Input}$ and $k_{Input} > {\sim}0.1$. For $N_{Input} = 1024$, this region disappears as the number of states in the cycle increases. We infer from this that, for large excesses of input, the

**Buffer regeneration module:**

$$\varnothing \xrightarrow{k_{\text{buf+}}} \text{Input}_{0,\text{buf}} \qquad\qquad \text{Input}_{0,\text{buf}} \xrightarrow{k_{\text{buf-}}} \varnothing$$

**Buffer conversion module:**

$$\text{Input}_{0,\text{cat}} + \text{Input}_{0,\text{buf}} \xrightarrow{k_{\text{cat}}} \text{Input}_{0,\text{cat}} + \text{Input}_0$$

**Figure 8.** Additional CRN modules for processing of catalytic inputs by our three-state FSA CRN example system. The buffer regeneration module maintains a stable and regenerating population of a buffered input species $\text{Input}_{i,\text{buf}}$ for each input species. The buffer conversion module uses the non-consumed (catalytic) input $\text{Input}_{i,\text{cat}}$ to catalyse conversion of $\text{Input}_{i,\text{buf}}$ into an 'activated' form $\text{Input}_i$ that can then be consumed by the reactions shown in figure 2 above.

system seems to be undertaking multiple partial state transitions and stabilizing on a final state that 'skips' one or more states out, driven by the fact that the higher flux through the transition mapping module is producing a larger number of flip molecules. In this framing, a successful state transition in this region is actually caused by the system skipping all the way round the cycle and ending up in the 'correct' ending state. Thus, for longer cycles, more skipping is required to reach a state that appears to be correct. This also explains the gulf between the regions of high and moderate success probabilities in the three-state system: this is presumably the region in which making two or three state transitions is most likely, but making four (to end up in the 'correct' state) is unlikely. In the 20-state system, this region would presumably be found with even higher values of the $k_{\text{Flip}}$ rate constant that we studied and/or higher amounts of input provided.

For $N_{\text{Input}} = 128$, however, this region appears to grow brighter as the number of states in the cycle increases from two to three to 20. This is probably due to states being 'skipped', and illustrates a key difference between the two-state system, compared with $n$-state systems for $n > 2$. In the two-state system, skipping a state brings the system back immediately into the original state, which provides additional impetus to the AM reactions that are initially attempting to keep the system in that state. Therefore, in this region of the weight space where skipping is seemingly relatively common, relatively low transition success probabilities are observed. However, once additional states are added, the ability to skip further ahead around the cycle of states serves to divide the population of created flip species between a larger number of states, as the lower amount of input means that fewer flip species will be produced overall. This means that the intended state transition (the first one, which will probably have the most flip species created for it) becomes more likely to succeed.

Our overall conclusion from this investigation is that the behaviour of our system is dependent on the choice of rate parameters; this is unsurprising, given that our design is a stochastic system that depends on competitive reactions for its function. Importantly, however, there is a clear region that gives 100% success rates. This work therefore justifies the set of rate constants chosen above ($k_{\text{Flip}} = 10$, $k_{\text{Input}} = 100$), as these fall within that region of highest success probability when $k_{AM} = 1$. We carried out this analysis for $n$-state cycles as they provide a simple yet extensible FSA architecture with which to test system behaviour; similar analyses for other FSA structures might yield slightly different results. However, we used these rate constant values for our subsequent work outlined below and continued to see reliable results, including in more complex FSA structures (figure 11).

## 2.4. Input processing reactions for non-consumed inputs

For practical purposes, we may not always want our FSA implementations to consume their input species. For example, a biochemical FSA deployed to observe temporal patterns of gene expression within a cell would need to observe intracellular concentrations of mRNAs, but consuming these to do so would significantly perturb the cell's internal metabolism, and this might even kill the cell. Therefore, it is desirable to implement a version of our system that does not consume the input species.

In this section, we introduce input processing modules that enable state transitions to be triggered on the rising edge of an input signal that is not, itself, consumed. Rather, those inputs catalyse the generation of consumable input that can be processed by the system outlined above. To do this, we augment our FSA CRN implementation with the reactions from the buffer regeneration and buffer conversion modules shown in figure 8. This system works by introducing an inactive 'buffered' input species $\text{Input}_{i,\text{buf}}$ corresponding to each actual input. These buffered inputs are generated at rate $k_{\text{buf+}}$ and degraded at

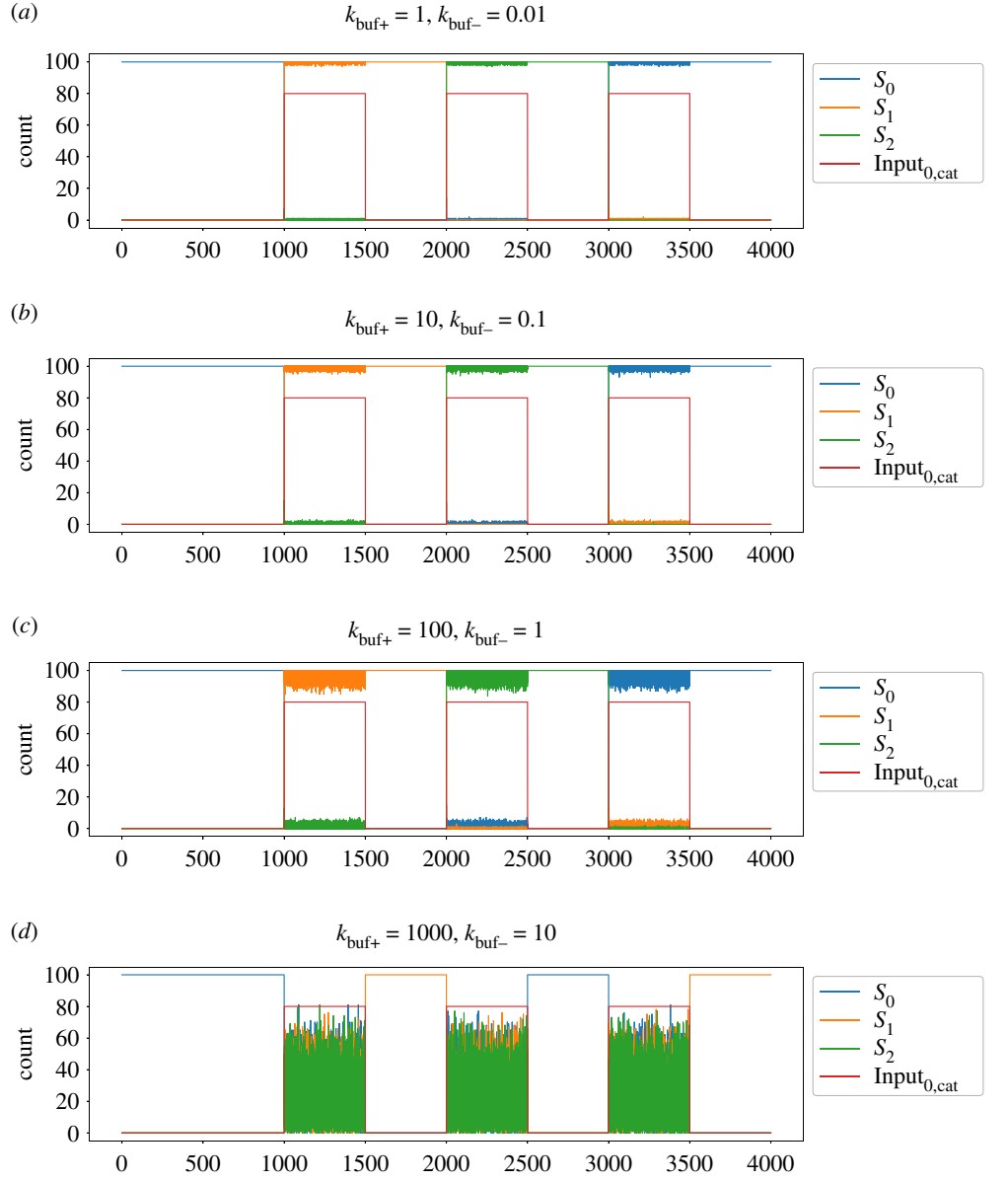

**Figure 9.** Timecourses for several state transitions of the FSA CRN presented in figure 2, augmented with the additional reactions from figure 8 to enable processing of non-consumed inputs. The rate constants used are $k_{AM} = 1$, $k_{Flip} = 10$, $k_{Input} = 100$, $k_{cat} = 10$ and $k_{FlipDecay} = 1$. The steady-state value for $Input_{0,buf}$ (not plotted) is $k_{buf+}/k_{buf-} = 100$ molecules, the input signals are a square wave switching between 0 and 80 molecules of $Input_{0,cat}$ present.

rate $k_{buf-}$, producing a steady state average count given by the ratio of the production and degradation rates, $k_{buf+}/k_{buf-}$. Conversion of 'buffered' inputs $Input_{i,buf}$ to active inputs $Input_i$ is catalysed by the non-consumed (i.e. catalytic) inputs $Input_{i,cat}$. This essentially converts the input into a spike of active inputs $Input_i$, without consuming the catalytic input $Input_{i,cat}$. This system requires that sufficient time has elapsed between presentation of successive catalytic inputs for the supply of buffered input species to regenerate itself. Note that, while an input signal is active, reactions from the buffer conversion module can proceed and thus create low-level noise in the state species, as the regenerating supply of inactive input is immediately converted to active input. Here, we exploit the noise-suppressing capabilities of the AM reactions to ensure that the system remains in a consistent state, and that the state representation is rapidly restored to a 100% representation when the input signal is removed.

Figure 9 presents example timecourses for three state changes in the three-state cycle CRN example introduced in figure 2, but adapted for non-consumed inputs. In each case, the system starts in state $S_0$ and successive presentations of the non-consumed input $Input_{0,cat}$ should transition the system to states $S_1$ then $S_2$, then back to $S_0$ again. Figure 9a–d shows the behaviour for four different sets of $k_{buf+}$ and $k_{buf-}$

parameters. In all four cases, the steady-state value for the input buffer species is $k_{buf+}/k_{buf-} = 100$; however, the absolute values of the rate constants differ. The $k_{buf+}$ rate constant represents the 'regeneration rate' of the buffer species; if it is too slow, the system cannot replace the buffer in time for the next input presentation. If it is too fast, however, we observe increasing levels of noise when the input signals are high, to the point that the states cannot be maintained reliably, leading to incorrect state transitions, as seen in figure 9d. This happens because the AM reactions are no longer happening fast enough to stabilize the state. In addition, too high a value for $k_{buf+}$ essentially amplifies the input signal, producing larger and larger spikes of the generated 'consumable' inputs Input$_i$, which can contribute to skipping through multiple states given a single presentation of input. These results do demonstrate that our FSA system can be modified to operate successfully without consuming the input signal species, which could be desirable for practical applications. We use this version in our example below.

## 2.5. Compiling deterministic finite automata into AM CRNs

We now summarize our algorithm for compiling an arbitrary deterministic finite state automaton into a corresponding CRN implementation. Formally, a deterministic FSA is a five-tuple $(Q, \Sigma, \delta, q_0, F)$ consisting of:

— a finite set of states, $Q$;
— a finite set of input symbols called the alphabet, $\Sigma$;
— a transition function $\delta: Q \times \Sigma \to Q$;
— an initial or start state $S_0 \in Q$; and
— a set of accepting states $F \subseteq Q$.

Our algorithm for compiling deterministic FSAs into stochastic AM CRNs is presented as pseudocode in figure 10. If there are $n$ distinct states $S_i \in Q$, the 'compileAM' function adds an $n$-way multi-state AM module covering the $n$ species $S_0, \ldots, S_{n-1}$, as outlined above. The 'compileTransitions' function adds reactions that implement the state transitions specified by the transition function $\delta: Q \times \Sigma \to Q$ given input signals drawn from $\Sigma$ and the current state. Each state and active input is mapped to an output state by the transition mapping module, and the transition is carried out by the transition module. To use the system with consumable inputs, the 'compileForConsumableInputs' function suffices to add the required reactions. To produce a system that *does not* consume its inputs, calling the 'compileForCatalyticInputs' function will also call the 'compileBufferRegeneration' and 'compileBufferConverston' functions to add the corresponding reactions to implement the buffer regeneration and conversion modules. The initial state is encoded in the initial state and inputs are introduced in order over time.

Finally, we note that the formal definition of a deterministic FSA also defines the set $F$ of accepting states $F$ as a subset of $Q$. In the interests of simplicity, our CRN implementation does not incorporate reactions into our FSA CRN system to distinguish accepting states from non-accepting states. In principle, this could be done by adding an additional species that is produced when the system enters an accepting state and is consumed when the system enters a non-accepting state. This is necessary because the FSA could pass through an intermediate accepting state but still end up in a non-accepting state when all input has been supplied. However, the inclusion of such additional reactants and products would not modify the overall design of our FSA CRNs in any interesting way, therefore, we omit them. In simulations, it is trivial to observe the state reached after all inputs have been supplied to determine if it is an accepting state or not. In a biochemical implementation, a reversible fluorescent reporter might be used to detect when the system is in an accepting state.

## 2.6. Four-bit square root example

To illustrate the use of our compilation process and its scalability, we present an FSA that calculates the floor of the square root of a four-bit binary number. This has become something of a classic problem in molecular computing research since it was solved using a DNA strand displacement reaction network by Qian & Winfree [14]. We defined an FSA over the alphabet {0, 1}, which goes into an accepting state when presented with a string at least four bits in length. There are four accepting states ($S_0$, $S_1$, $S_2$ and $S_3$), corresponding to the four possible results of a four-bit square root operation. The accepting state entered by the FSA thus records the square root of the binary number represented by the first four bits in the sequence. We also added a Reset symbol so that the system can be returned to its initial state at any time.

```
define states = [S_0, ..., S_i]
define transitions = [(Input, Initstate, FinalState)_0,... (Input, InitState, FinalState)_j]

function compileAM(states):
        for S_i in states:
                CRN.append(S_i + B ──k_AM──→ S_i + S_i)
                for S_j in states:
                        if i < j:
                                CRN.append(S_i + S_i ──k_AM──→ B + B)

function compileTransitions(transitions) :
        for (Input_x, S_i, S_f) in transitions :
                CRN.append(Input_x + S_i ──k_input──→ Flip_{i→f} + S_i)
                CRN.append(Flip_{i→f} + S_i ──k_flip──→ S_f)

function compileBufferRegeneration(transitions):
        for (Input_x, _, _) in transitions:
                CRN.append(∅ ──k_{buf+}──→ Input_{x,buf})
                CRN.append(Input_{x,buf} ──k_{buf−}──→ ∅)

function compileBufferConversion(transitions):
        for (Input_x, _, _) in transitions:
                CRN.append(Input_{x,cat} + Input_{x,buf} ──k_cat──→ Input_{x,cat} + Input_x)

function compileForConsumableInputes(states, transitions):
        compileAM(states)
        compileTransitions(transitions)

function compileForCatalyticInputes(states, transitions):
        compileAM(states)
        compileTransitions(transitions)
        compileBufferRegeneration(transitions)
        compileBufferConversion(transitions)
```

**Figure 10.** Pseudocode algorithm for compilation of an arbitrary FSA to one of our AM-based stochastic CRN implementations. The user will create an empty CRN, define a set of states and transitions, and then use either the 'compileForConsumableInputs' or 'compileForCatalyticInputs' function, depending on the desired interface.

We constructed this CRN by first enumerating the binary tree of all possible four-bit input strings, as shown in figure 11. This was done by first forming a binary decision tree for the four-bit square root function, identifying those bit-strings that produce the same square root value, and using this information to compress this binary tree into an FSA. For clarity, figure 11 omits the self-edges on the four accepting states and the edges that represent the transitions from any state back to the starting state (Init) on the Reset input signal. The latter enable the system to be re-used to compute multiple square roots sequentially.

The FSA from figure 11 was compiled into a stochastic CRN using the algorithm from figure 10 to produce a CRN system that accepts catalytic inputs. We then simulated the system to compute the square roots of 10 (binary 1010) followed by 7 (binary 0111), by supplying the sequence of inputs:

$$Input_1, Input_0, Input_1, Input_0, Reset, Input_0, Input_1, Input_1, Input_1, Reset.$$

The resulting timecourse from the stochastic simulation is presented in figure 12. This trace demonstrates the system successfully reaching the accepting state $S_3$ representing the result that $\lfloor\sqrt{10}\rfloor = 3$, being reset, and subsequently reaching the accepting state $S_2$ to show that $\lfloor\sqrt{7}\rfloor = 2$. This illustrates that our system can successfully compute over multiple rounds of input. Furthermore, this FSA has 11 states, which

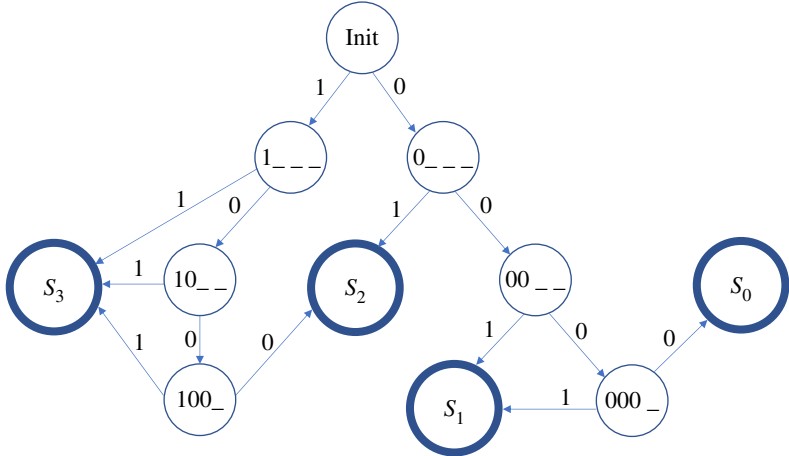

**Figure 11.** A finite state automaton that calculates the square root of a four-bit binary number. Not shown are self edges for inputs 1 and 0 at each accepting state (nodes with thicker outlines), as well as a third Reset input that will take any state back to the Init state.

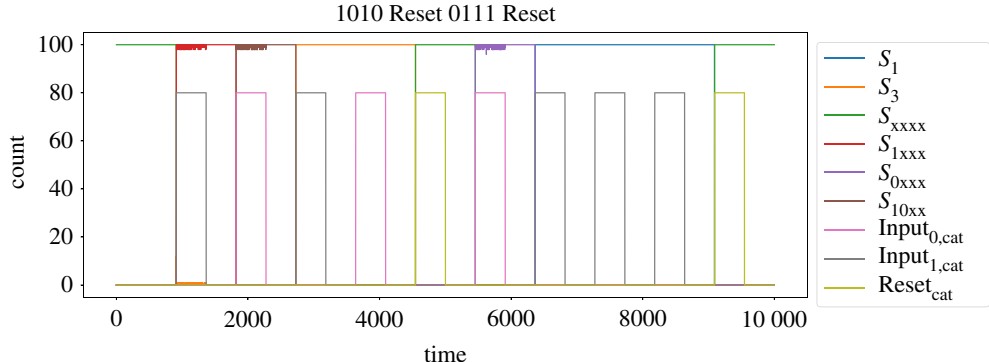

**Figure 12.** Timecourse of the square root FSA in figure 11 calculating the binary square roots of 10 and 7. The first four inputs are catalytic inputs that spell out '1010' in binary, producing the response that $\lfloor\sqrt{10}\rfloor = 3$ by transitioning to the accepting state $S_3$. A pulse of the Reset species returns the system to its initial state, and four further inputs spell out '0111' in binary, producing the response that $\lfloor\sqrt{7}\rfloor = 2$ by transitioning to the accepting state $S_2$. Note that the FSA enters an accepting state as soon as it receives enough inputs to know the correct answer for sure. Species representing the non-accepting states, input buffers and activated inputs are not plotted in this timecourse. Rate constants used for this simulation were $k_{AM} = 1$, $k_{Flip} = 10$, $k_{Input} = 100$, $k_{FlipDecay} = 1$, $k_{buf+} = 1$, $k_{buf-} = 0.01$ and $k_{cat} = 10$. All input signals are presented as square waves switching between 0 and 80 molecules.

demonstrates the potential for scale-up of the multi-stable AM memory element to store information robustly, even in a stochastic system with relatively few copies of individual species.

# 3. Discussion

Biomolecular devices live in the nanoscale realm of stochastic interactions, and as such they require careful engineering to ensure predictable behaviour that is robust to stochastic fluctuations. We have tackled this challenge in the context of deterministic FSAs to define a class of stochastic CRN implementations that can robustly store state information and transition between states in the face of stochastic noise. In our design, maintaining a coherent global chemical representation of the state of the system is a key problem. Our implementation represents the individual states of an FSA as a set of states of the underlying stochastic chemical implementation in which one species is dominant: either accounting for 100% of the state species or sufficiently close to it (the latter in implementations where there may be some noise in the state species signals). This is achieved by using the approximation majority (AM) distributed consensus algorithm [10], which drives the system toward an equilibrium where one of two competing species dominates, based on whichever was initially in the majority. These stable global states represent one of the abstract states of the FSA. We then

developed a simple mechanism for switching states, either via spikes of input or processing of catalytic inputs. We showed an algorithm for compiling an arbitrary deterministic FSA into an FSA CRN that functions robustly in the stochastic regime and demonstrated the use of this system to simulate a non-trivial FSA that solves the four-bit binary square root problem. This demonstrates the potential for our system to implement finite-state computations robustly while using relatively low molecular counts, e.g. as control systems for bioinspired robots [15].

## 3.1. Related work

Previously proposed molecular computing systems have addressed the problem of maintaining a globally coherent representation of the current state by representing the state using a single copy of a state molecule [6,7]. However, many systems of interest to synthetic biologists and biomedical engineers, such as *E. coli* cells or engineered synthetic cells, often involve of the order of hundreds or thousands (or sometimes just tens) of copies of a given species. Nevertheless, decision-making must be coordinated robustly across the entire extent of the system. Systems at this scale are significantly influenced by stochastic noise due to the relatively large variance in localized species concentrations. As the scale grows, noise has a smaller effect on the relative species counts in a chemical reaction network, and their behaviour can be modelled deterministically using ordinary differential equations (ODEs) as a function of the stoichiometry and relative rate constants in the CRN.

Previous work on the AM system has shown that it occurs in a number of cellular decision-making networks [3]. Other proposed synthetic system designs that require the property of accurate determination of the initial majority have employed the AM algorithm as a digital switch to maintain a globally consistent representation of the overall state [16,17]. Here, we present a transition species that converts one state species to another, whose production is catalysed by the current state species in the presence of an input species.

Perhaps the closest related work to ours is that of Klinge *et al.* [18] on implementations of robust FSAs in deterministic CRNs. That work uses an ODE model as opposed to our stochastic one and proves theoretical results on robustness, whereas we opt instead for empirical characterization of behaviour under various sets of reaction rate parameters. Their system uses multiple sequential presentations of input signals to prime the system for the next input symbol and to reset it for the subsequent round, whereas ours just requires a single presentation of the input symbol alone at each step. Thus, in this sense, our system may be seen to be simpler for use in practical applications. Finally, their system is non-deterministic in the sense that it can be interpreted as exploring all possible states in parallel. Our approach could, in principle, also implement non-deterministic systems, as discussed below. In addition, Oishi & Klavins reported an implementation of finite state machines in genetic regulatory networks [19]. That system uses multiple distinct repressors to keep one state dominant; in our system, that role is carried out by the AM reactions.

## 3.2. Possibility of wet chemistry implementations

Previous experimental work reported a DNA strand displacement-based implementation of the approximate majority algorithm [11], thereby illustrating a possible route to wet laboratory implementations of the algorithms that we study in this research. In addition to the AM component, the other control reactions that make up our FSA CRNs could also be implemented using DNA strand displacement [9]. That paper shows that arbitrary abstract chemical reaction networks can be directly implemented using strand displacement networks, thereby providing a direct route from our design to a laboratory implementation, at least in principle. This approach would require careful sequence design to ensure that the system adheres to the reaction rules specified here.

With regard to the required tuning of rate constants, toehold lengths can be used to control strand displacement reaction rates over several orders of magnitude [20]. This approach has previously been used to generate large discrepancies in strand displacement reaction rates for practical use in DNA-based molecular 'timer' circuits [21]. Alternative tuning approaches include the use of 'buffers' to enable effective reaction rates to be tuned by controlling the available quantity of an additional reactant species [22,23]. Other approaches to molecular computing have previously been used to implement similar automata, including restriction enzyme-based systems [24].

An interesting aspect of our system is the scalability of the number of molecules required to store global state information robustly. Our square root example FSA (figure 11) shows that a reasonably complex FSA can be implemented in a stochastic system that uses just 100 molecules to store the state

information. Our use of the AM algorithm to regulate state transitions and to ensure state robustness means that the complexity of the FSA implementation is found in the CRN instead, whose number of reactions is quadratic in the number of distinct state species. For example, our square root FSA CRN involves 174 abstract reactions and 50 abstract species (all reactions are bimolecular at most). The number of individual molecules required to store the state would probably only be an issue in the face of noise that degrades the state by removing or flipping individual state molecules.

## 3.3. Extension to non-deterministic automata

Non-deterministic finite automata are automata in which the rule-set $\delta$ is a relation as opposed to a function. This means that there may be more than one possible next state given the current state and the supplied input symbol. There are several interpretations of this, including that all possible next states are somehow tried in parallel (as explored by Klinge et al. [18]), as is commonly assumed in theoretical computer science studies of non-deterministic automata. An alternative interpretation would be for the system to commit completely to one of the possible transitions. This would have parallels to seemingly random behaviour observed in certain single-celled organisms [2]. Indeed, there has been some previous work on the latter in CRNs [25], although that work involved a single molecule undergoing fast reaction to make the choice according to specified probabilities, which is then 'locked in'.

## 3.4. Possible applications

This work is intended to provide a general framework for compiling a high-level description of automata into abstract chemical reaction networks, forming a basis for creative implementation in various chemical systems. Such mechanisms of programmable control over biomolecular systems would enable new technology in medicine and synthetic biology. Potential applications for the chemical deterministic FSA presented here could include implementation as a DNA computing system for medical diagnostics [26], in which the automaton can assume a readable state via fluorescent reporters, indicating that it has encountered a specific sequence of biomolecular inputs. Future applications might include chemical nanofactories that may benefit from finite state control of nanorobots [27], in which the engineer can define a sequence of programmed actions resulting in synthesis of biomolecules.

Data accessibility. Code for the ProBioSim simulator used in this work is available at: https://github.com/matthewlakin/ProBioSim. Other code for the creation of models and analysis of data is available as electronic supplementary material.
Competing interests. We declare we have no competing interests.
Funding. This material is based upon work supported by the National Science Foundation under grant nos. 1518861 and 1525553.

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
