## [Peer Review File · Royal Society Open Science]

Review History

RSOS-211310.R0 (Original submission)

Review form: Reviewer 1

Is the manuscript scientifically sound in its present form?

Yes

Are the interpretations and conclusions justified by the results?

Yes

Is the language acceptable?

Yes

Do you have any ethical concerns with this paper?

No

Have you any concerns about statistical analyses in this paper?

No

Recommendation?

Accept with minor revision (please list in comments)

Comments to the Author(s)

In the paper, the authors demonstrated the principles and modules needed for building finite state machines with chemical reaction networks. The paper is nicely written and is advised to be published with following suggestions:

The authors first introduced the major logic that enables the implementation of finite state machines - the approximate majority algorithm, where the purpose of such an algorithm is to allow the system to converge to a single state. While the principle works flawlessly in-silico, one minor comment here, as this is a computational paper, is the difficulty in actual implementation with chemical or biochemical molecules. One may need to think carefully about how to design reversible molecules that would react following the proposed rules. It would thus help if the authors may provide some thoughts on the implementation if possible.

The second key module the authors implemented for building finite state machines is the transition mapping and introduction of "flip species". The authors demonstrated how the introduction of new species stabilizes the state transitions of finite state machines by showing the two cases of simulation with and without the implementation of "flip species" - with the non-inclusion case erring with failed state transition. While this is a straightforward example, the stochasticity of the system may render this result difficult to reproduce. It would thus greatly help readers to visualize the noise and show the stabilization effect for running the two designs with multiple runs, then use the derived mean and standard deviation to visualize stability effect of flip species introduction provides, thus, a major comment here would be the introduction of such a figure added in figure 3. Also, it is not clear how the simulations of the non-flip-species-cases were conducted, this is essential to include for readers' reference.

In the following sections, the authors performed detailed parametric studies and characterization to learn how different rates affect the state-switching robustness of the finite state machines. The heatmaps offer clear visualization to learn the trend and the authors also provide thoughtful interpretations. However, another major comment here is that how the authors are evaluating the FSM correctness is unclear, and should be carefully explained. It is clear to interpret 100% and 0% correctness, but there are multiple regions with values lying in-between and is confusing for readers - for example, did the FSMs skip some states or did they fail to skip states? The authors should clarify the definition in the main text or in the captions below.

Finally, the implementation of solving the square root problem is a great and interesting example demonstrating the power of useful finite state machines with CRN implementations!

Review form: Reviewer 2

Is the manuscript scientifically sound in its present form?

Yes

Are the interpretations and conclusions justified by the results?

Yes

Is the language acceptable?

Yes

Do you have any ethical concerns with this paper?

No

Have you any concerns about statistical analyses in this paper?

No

Recommendation?

Accept as is

Comments to the Author(s)

The authors present a scheme to compile arbitrary Finite State Automata (FSA) into stochastic Chemical Reaction networks (CRN). They do this by using Approximate Majority to stabilize the state of the system after partial transitions are triggered by an Input. This makes the construction robust to stochastic noise. Their scheme is pithy as the number of species and reactions scale linearly and quadratically respectively, with the number of states in the FSA. They also present a variant of their scheme where the input species are not consumed, and instead catalytically trigger a cascade of transitions in a buffer system, which in turn mediates the actual transitions in the FSA. The problem is an important one to solve, as FSA are basic building blocks of computation. Understanding how to implement FSA by reaction networks allows bringing the computational lens to an examination of chemical computing, including processes in living cells.

The proposed scheme is (unsurprisingly) sensitive to the choice of reaction rates, which the authors empirically characterize through simulations. They have considered the various ways their compiled CRN could fail and have then tuned the reaction rates to avoid these. This method of setting the rates wouldn't scale well with the complexity of the FSA. However the authors do mention that picking rates $K_{AM} = 1$, $K_{Flip} = 10$ and $K_{input} = 100$ scaled well in their experience. It would be nicer if they mathematically claimed and proved this.

Finally, they demonstrate their construction by compiling a FSA for accepting the square root of a 4-bit number into a CRN via their scheme, which are the kind of things that are always a delight to see.

The construction is simple yet very interesting. It is also pithy as the number of species only scales linearly and the number of reactions only scales quadratically.

To summarize, this is a simple but important advance. I imagine lots of papers will build on this work in future, and strongly recommend publication.

Decision letter (RSOS-211310.R0)

Dear Dr Lakin

On behalf of the Editors, we are pleased to inform you that your Manuscript RSOS-211310 "Robust finite automata in stochastic chemical reaction networks" has been accepted for publication in Royal Society Open Science subject to minor revision in accordance with the referees' reports. Please find the referees' comments along with any feedback from the Editors below my signature.

Please submit your revised manuscript and required files (see below) no later than 7 days from today's (ie 10-Nov-2021) date. Note: the ScholarOne system will 'lock' if submission of the revision is attempted 7 or more days after the deadline. If you do not think you will be able to meet this deadline please contact the editorial office immediately.

on behalf of Professor Ion Petre (Associate Editor) and Marta Kwiatkowska (Subject Editor)
openscience@royalsociety.org

Associate Editor Comments to Author (Professor Ion Petre):

Comments to the Author:

The reviewers offer a few recommendations to improve the paper. Please consider them carefully and respond to their suggestions point by point.

Reviewer comments to Author:

Reviewer: 1

Comments to the Author(s)

In the paper, the authors demonstrated the principles and modules needed for building finite state machines with chemical reaction networks. The paper is nicely written and is advised to be published with following suggestions:

The authors first introduced the major logic that enables the implementation of finite state machines - the approximate majority algorithm, where the purpose of such an algorithm is to allow the system to converge to a single state. While the principle works flawlessly in-silico, one minor comment here, as this is a computational paper, is the difficulty in actual implementation with chemical or biochemical molecules. One may need to think carefully about how to design reversible molecules that would react following the proposed rules. It would thus help if the authors may provide some thoughts on the implementation if possible.

The second key module the authors implemented for building finite state machines is the transition mapping and introduction of "flip species". The authors demonstrated how the introduction of new species stabilizes the state transitions of finite state machines by showing the two cases of simulation with and without the implementation of "flip species" - with the non-inclusion case erring with failed state transition. While this is a straightforward example, the

stochasticity of the system may render this result difficult to reproduce. It would thus greatly help readers to visualize the noise and show the stabilization effect for running the two designs with multiple runs, then use the derived mean and standard deviation to visualize stability effect of flip species introduction provides, thus, a major comment here would be the introduction of such a figure added in figure 3. Also, it is not clear how the simulations of the non-flip-species-cases were conducted, this is essential to include for readers' reference.

In the following sections, the authors performed detailed parametric studies and characterization to learn how different rates affect the state-switching robustness of the finite state machines. The heatmaps offer clear visualization to learn the trend and the authors also provide thoughtful interpretations. However, another major comment here is that how the authors are evaluating the FSM correctness is unclear, and should be carefully explained. It is clear to interpret 100% and 0% correctness, but there are multiple regions with values lying in-between and is confusing for readers - for example, did the FSMs skip some states or did they fail to skip states? The authors should clarify the definition in the main text or in the captions below.

Finally, the implementation of solving the square root problem is a great and interesting example demonstrating the power of useful finite state machines with CRN implementations!

Reviewer: 2

Comments to the Author(s)

The authors present a scheme to compile arbitrary Finite State Automata (FSA) into stochastic Chemical Reaction networks (CRN). They do this by using Approximate Majority to stabilize the state of the system after partial transitions are triggered by an Input. This makes the construction robust to stochastic noise. Their scheme is pithy as the number of species and reactions scale linearly and quadratically respectively, with the number of states in the FSA. They also present a variant of their scheme where the input species are not consumed, and instead catalytically trigger a cascade of transitions in a buffer system, which in turn mediates the actual transitions in the FSA. The problem is an important one to solve, as FSA are basic building blocks of computation. Understanding how to implement FSA by reaction networks allows bringing the computational lens to an examination of chemical computing, including processes in living cells.

The proposed scheme is (unsurprisingly) sensitive to the choice of reaction rates, which the authors empirically characterize through simulations. They have considered the various ways their compiled CRN could fail and have then tuned the reaction rates to avoid these. This method of setting the rates wouldn't scale well with the complexity of the FSA. However the authors do mention that picking rates $K_{AM} = 1$, $K_{Flip} = 10$ and $K_{input} = 100$ scaled well in their experience. It would be nicer if they mathematically claimed and proved this.

Finally, they demonstrate their construction by compiling a FSA for accepting the square root of a 4-bit number into a CRN via their scheme, which are the kind of things that are always a delight to see.

The construction is simple yet very interesting. It is also pithy as the number of species only scales linearly and the number of reactions only scales quadratically.

To summarize, this is a simple but important advance. I imagine lots of papers will build on this work in future, and strongly recommend publication.

===PREPARING YOUR MANUSCRIPT===

one version should clearly identify all the changes that have been made (for instance, in coloured highlight, in bold text, or tracked changes);

===PREPARING YOUR REVISION IN SCHOLARONE===

-- If you are requesting an article processing charge waiver, you must select the relevant waiver option (if requesting a discretionary waiver, the form should have been uploaded, see 'File upload' above).

-- If you have uploaded any electronic supplementary (ESM) files, please ensure you follow the guidance at <https://royalsociety.org/journals/authors/author-guidelines/#supplementary-material> to include a suitable title and informative caption. An example of appropriate titling and captioning may be found at https://figshare.com/articles/Table_S2_from_Is_there_a_trade-off_between_peak_performance_and_performance_breadth_across_temperatures_for_aerobic_scope_in_teleost_fishes_/3843624.

Author's Response to Decision Letter for (RSOS-211310.R0)

See Appendix A.

Decision letter (RSOS-211310.R1)

Dear Dr Lakin,

I am pleased to inform you that your manuscript entitled "Robust finite automata in stochastic chemical reaction networks" is now accepted for publication in Royal Society Open Science.

on behalf of Professor Ion Petre (Associate Editor) and Marta Kwiatkowska (Subject Editor)
openscience@royalsociety.org

November 12, 2021

Dear Editor,

This letter supports the resubmission of our revised manuscript entitled “Robust finite automata in stochastic chemical reaction networks” to Royal Society Open Science.

We thank the reviewers for their positive response to the initial version of our manuscript. In preparing this revision, we have addressed the points raised in their reviews, as outlined below.

Reviewer: 1

Comments to the Author(s)

In the paper, the authors demonstrated the principles and modules needed for building finite state machines with chemical reaction networks. The paper is nicely written and is advised to be published with following suggestions:

The authors first introduced the major logic that enables the implementation of finite state machines - the approximate majority algorithm, where the purpose of such an algorithm is to allow the system to converge to a single state. While the principle works flawlessly in-silico, one minor comment here, as this is a computational paper, is the difficulty in actual implementation with chemical or biochemical molecules. One may need to think carefully about how to design reversible molecules that would react following the proposed rules. It would thus help if the authors may provide some thoughts on the implementation if possible.

We thank the reviewer for their positive recommendation of our manuscript. We have addressed this question of implementability in the Discussion (subsection entitled “Possibility of wet chemistry implementations”) by adding the following text on how our design could be converted into a DNA strand displacement-based implementation: *“That paper shows that arbitrary abstract chemical reaction networks can be directly implemented using strand displacement networks, thereby providing a direct route from our design to a laboratory implementation, at least in principle. This approach would require careful sequence design to ensure that the system adheres to the reaction rules specified here.”*

The second key module the authors implemented for building finite state machines is the transition mapping and introduction of “flip species”. The authors demonstrated how the introduction of new species stabilizes the state transitions of finite state machines by showing the two cases of simulation with and without the implementation of “flip species” - with the non-inclusion case erring with failed state transition. While this is a straightforward example, the stochasticity of the system may render this result difficult to reproduce. It would thus greatly help readers to visualize the noise and show the stabilization effect for running the two designs with multiple runs, then use the derived mean and standard deviation to visualize stability effect of flip species introduction provides, thus, a major comment here would be the introduction of such a figure added in figure 3.

This is an excellent suggestion and we have included precisely these plots as Figure 3c and 3d in the revised version of the manuscript. As expected, these show tight error bars around the correct sequence of state transitions when the flip species are used and broad error bars and indeterminate average states (due to incorrect transitions into different states in each trace) in the averaged system when the flip species are *not* used. This addition demonstrates more convincing statistical evidence for the utility of the flip species approach and we thank the review for suggesting it.

Also, it is not clear how the simulations of the non-flip-species-cases were conducted, this is essential to include for readers' reference.

This omission has been corrected by clarifying the explanation of the CRN that was used for the "no flip" simulations, by adding the following text: *"In other words, in the "no flip" case, the transition mapping and transition modules from Figure 2 are replaced with the following three reactions"*, followed by a listing of the corresponding three reactions.

In the following sections, the authors performed detailed parametric studies and characterization to learn how different rates affect the state-switching robustness of the finite state machines. The heatmaps offer clear visualization to learn the trend and the authors also provide thoughtful interpretations. However, another major comment here is that how the authors are evaluating the FSM correctness is unclear, and should be carefully explained. It is clear to interpret 100% and 0% correctness, but there are multiple regions with values lying in-between and is confusing for readers - for example, did the FSMs skip some states or did they fail to skip states? The authors should clarify the definition in the main text or in the captions below.

We agree that we did not explain clearly how correct state transitions are detected in the simulation. To address this, we added the following text to the relevant part of the manuscript: *"Correct state transitions were detected by averaging the species counts from the middle fifty percent of time points occurring between two perturbations. For a given time point, if the count of the relevant state species is greater than 80% of the total number of state species (which in this case means 80 copies), then we say the system is in the corresponding state."*

Finally, the implementation of solving the square root problem is a great and interesting example demonstrating the power of useful finite state machines with CRN implementations!

We are glad that the reviewer appreciated this nod to molecular computing history.

Reviewer: 2

Comments to the Author(s)

The authors present a scheme to compile arbitrary Finite State Automata (FSA) into stochastic Chemical Reaction networks (CRN). They do this by using Approximate Majority to stabilize the state of the system after partial transitions are triggered by an Input. This makes the construction robust to stochastic noise. Their scheme is pithy as the number of species and reactions scale linearly and quadratically respectively, with the number of states in the FSA. They also present a variant of their scheme where the input species are not consumed, and instead catalytically trigger a cascade of transitions in a buffer system, which in turn mediates the actual transitions in the FSA. The problem is an important one to solve, as FSA are basic building blocks of computation. Understanding how to implement FSA by reaction networks allows bringing the computational lens to an examination of chemical computing, including processes in living cells.

We appreciate the positive comments on the scalability of the system design in terms of the numbers of species and reactions.

The proposed scheme is (unsurprisingly) sensitive to the choice of reaction rates, which the authors empirically characterize through simulations. They have considered the various ways their compiled CRN could fail and have then tuned the reaction rates to avoid these. This method of setting the rates wouldn't scale well with the complexity of the FSA. However the authors do mention that picking rates $K_{AM} = 1$, $K_{Flip} = 10$ and $K_{input} = 100$ scaled well in their experience. It would be nicer if they mathematically claimed and proved this.

While it is true that the performance of our stochastic implementations of finite state automata are dependent on the rate constant values, we note that the question of scaling with complexity is not a significant barrier to the application of our technique since the number of rates required does not increase with increasing size of the implemented FSA. Moreover, although we do observe slightly different "performance heatmaps" for the n -state cycle FSA examples, there is a reasonable degree of overlap which means that rate constants are transferrable between different FSA designs to a significant extent. For example, the rate constants used in the square root FSA implementation example were not optimized for that specific example but rather transferred from an earlier system. As a general rule of thumb, increasing the separation of timescales between rates tends to improve accuracy of the FSA state transitions. With regard to mathematical proof, we note that even for 100 copies of each molecule, the state space for the system will be far too large to admit analytical solutions for system accuracy.

Finally, they demonstrate their construction by compiling a FSA for accepting the square root of a 4-bit number into a CRN via their scheme, which are the kind of things that are always a delight to see.

The construction is simple yet very interesting. It is also pithy as the number of species only scales linearly and the number of reactions only scales quadratically.

To summarize, this is a simple but important advance. I imagine lots of papers will build on this work in future, and strongly recommend publication.

We thank the reviewer for their positive recommendation.

We thank you in advance for your consideration of our revised manuscript.

Yours faithfully,

Matthew Lakin

mlakin@cs.unm.edu